# Potential of Ca-Complexed in Amino Acid in Attenuating Salt Stress in Sour Passion Fruit Seedlings

**DOI:** 10.3390/plants13202912

**Published:** 2024-10-17

**Authors:** Antônio Gustavo de Luna Souto, Angela Maria dos Santos Pessoa, Sarah Alencar de Sá, Nayana Rodrigues de Sousa, Emerson Serafim Barros, Francimar Maik da Silva Morais, Fagner Nogueira Ferreira, Wedson Aleff Oliveira da Silva, Rafael Oliveira Batista, Daniel Valadão Silva, Rita Magally Oliveira da Silva Marcelino, Hans Raj Gheyi, Geovani Soares de Lima, Rosa Maria dos Santos Pessoa, Mailson Monteiro do Rêgo

**Affiliations:** 1Universidade Federal Rural do Semi-Árido, Mossoro 59625-900, RN, Brazil; angelapessoapb@gmail.com (A.M.d.S.P.); sarahalencardesa28@gmail.com (S.A.d.S.); nayanasousa12@hotmail.com (N.R.d.S.); emerson.serafim.barros@gmail.com (E.S.B.); francimar.morais@alunos.ufersa.edu.br (F.M.d.S.M.); fagnernf@gmail.com (F.N.F.); wedson.silva@alunos.ufersa.edu.br (W.A.O.d.S.); rafaelbatista@ufersa.edu.br (R.O.B.); daniel.valadao@ufersa.edu.br (D.V.S.); rm.magally@gmail.com (R.M.O.d.S.M.); 2Programa de Pós-Graduação em Engenharia Agrícola, Universidade Federal de Campina Grande, Campina Grande 58429-900, PB, Brazil; hans.gheyi@ufcg.edu.br (H.R.G.); geovani.soares@professor.ufcg.edu.br (G.S.d.L.); 3Centro de Ciências Agrárias, Universidade Federal da Paraíba, Areia 58397-000, PB, Brazil; rosapessoaa@gmail.com (R.M.d.S.P.); mailson@cca.ufpb.br (M.M.d.R.)

**Keywords:** *Passiflora edulis* Sims, sustainability, water salinity, chelated calcium, gas exchange

## Abstract

Salt stress results in physiological changes that inhibit plant growth and development. Ca-complex sources are used as a potential salt stress attenuator. This study was carried out with the aim of verifying the effects of Ca-complex sources in reducing the effects of saline water stress on the physiological aspects of sour passion fruit seedlings. The experiment was carried out in a randomized block design with a 2 × 2 × 3 factorial scheme, consisting of two cultivars of sour passion fruit (BRS GA1 and BRS SC1), two levels of water salinity (electrical conductivity of 0.5 and 4.0 dS m^−1^) and three sources of Ca-complex (without, organic acids and amino acids). The traits measured at 60 days after sowing were gas exchange, chlorophyll indices, chlorophyll fluorescence, electrolyte leakage, and relative water content in the leaf limb. Under moderate water salinity, the application of Ca-complex in amino acids promoted increases of 49.84% and 43.71%, respectively, in the efficiency of water use and carboxylation. The application of complex sources increased the stability of cell membranes, reducing electrolyte leakage, providing higher relative water content in seedlings irrigated with moderately saline water. From the results, we conclude that Ca-complex sources have potential as modulators of moderately saline water stress in sour passion fruit seedlings.

## 1. Introduction

Increased soil salinity due to irrigation with water containing high concentrations of salts is one of the main obstacles to agricultural expansion and food production in various regions of the world [1]. The accumulation of salts in the topsoil near plant roots causes loss of fertility and changes in the soil’s structural properties, resulting in significant negative impacts on crop yields [2]. This problem is particularly relevant in arid and semi-arid climate regions, where low rainfall, high temperatures and high evaporation rates exacerbate the effects of salinization [3].

In Brazil, the occurrence of saltwater is a persistent challenge, especially in the semi-arid areas of the northeast [4,5]. Many farmers in these regions depend exclusively on this type of water for their agricultural activities [6]. The high concentration of salts creates a pressure gradient in the soil, which limits the absorption of water by plants and implies deleterious effects on physiological functioning and nutritional status, mainly due to the presence of toxic ions (Na^+^ and Cl^−^) in plant tissues that reduce the capacity for plant growth and development [7,8].

The sour passion fruit (*Passiflora edulis* Sims), a species classified as sensitive to salt stress [9], shows a reduction in chlorophyll fluorescence, gas exchange and increased cell damage through greater electrolyte leakage and lower relative water content, affecting the growth and quality of seedlings during the vegetative phase [4,10]. However, the intensity with which salt stress affects plants depends on a number of factors, such as the variation in tolerance levels between passion fruit cultivars, soil and climate conditions, cultural treatments and irrigation and fertilization management [4,6].

Calcium (Ca^2+^), a secondary macro nutrient for plants, acts in metabolism through molecular signaling and plays an active role in mineral nutrition and plant growth, as well as being a structural constituent of the cell wall [11]. Ca^2+^ is also considered crucial in altering selective ion uptake and increasing plant tolerance to salt stress [8,12]. The role of this macro nutrient in salt stress tolerance is related to the accumulation of the element in cells or intracellular release as a way of signaling plants to adjust their physiological functions under stress conditions [13,14].

Calcium is an immobile element in the phloem, being transported exclusively via the xylem by transpiratory flow. It acts in the rhizosphere, reducing water availability, affecting leaf transpiration due to the stimulus of the partial closure of stomata, which can aggravate calcium deficiency in plant tissues [15]. The use of chelating or complexing agents in agriculture aims to protect metal or non-metal ions from undesirable chemical reactions and improve their availability for plant root absorption, including calcium ions [16,17]. Among natural complexing agents, such as amino acids and organic acids, each act differently on the bioavailability of nutrients for plant absorption [17,18,19].

The sustainability of agriculture and food security depends on research efforts aimed at selecting varieties that are better adapted to saline growing conditions and developing management practices that minimize the effects of salt stress on plants [20]. Therefore, our hypothesis is that the use of Ca-complexed sources can increase calcium mobility in plants, promoting the modulation of the physiological mechanisms of acclimatization of sour passion fruit seedlings via the supply of Ca^2+^ as a secondary messenger under conditions of salt stress and that this response may vary between different genetic materials of sour passion fruit. Therefore, the aim of this study was to verify the potential of complexed calcium sources in reducing the effects of water salinity on physiological aspects in seedlings of sour passion fruit cultivars in a semi-arid area.

## 2. Results

### 2.1. Gas Exchange and Photosynthetic Efficiency

According to the results in Table 1, the interaction among cultivars × water salinity × calcium sources influenced the rate of net photosynthesis and the water use efficiency and instantaneous carboxylation efficiency. Passion fruit leaf transpiration responded to the interaction cultivars × water salinity and cultivars × calcium sources, while stomatal conductance was influenced by the interaction between water salinity and calcium sources. In isolation, water salinity interfered with internal CO_2_ concentration and instantaneous water use efficiency, while complex calcium sources had a significant effect on the leaf Ci of seedlings.

The rate of net photosynthesis showed a significant difference between the cultivars, with BRS GA1 standing out, which was superior in the treatments under irrigation with 4.0 dS m^−1^ water, especially when associated with the application of calcium complexed with amino acids (Figure 1A). Irrigation with 4.0 dS m^−1^ water reduced the photosynthetic rate of the seedlings in most of the treatments with different calcium sources. However, an exception was observed in the BRS GA1 cultivar treated with calcium complexed with amino acids, where there was no significant difference between the salinities of the irrigation water.

The application of calcium complexed with amino acids (Ca-AA) promoted a higher rate of photosynthesis in the BRS GA1 cultivar irrigated with moderately saline water, showing an increase of 37.96 and 55.07%, respectively, compared to the treatments without calcium and with calcium complexed in organic acids (Figure 1A).

The water use efficiency (WUE) values of seedlings did not differ significantly between the BRS GA1 and BRS SC1 cultivars when irrigated with saline water and the application of Ca-complex sources (Figure 1B). In the BRS GA1 cultivar, WUE decreased when irrigated with moderately saline water, except in those treated with Ca-complex in amino acids, which did not differ statistically from the mean value of seedlings irrigated with low salinity water. On the other hand, cv. BRS SC1 showed a reduction in WUE when irrigated with moderately saline water, regardless of the application of complexed calcium sources. The losses were 23.98, 34.88 and 36.59%, respectively, for plants in the treatments without calcium (W-Ca), calcium complexed with organic acids (Ca-OA) and calcium complexed with amino acids (Ca-AA). The application of calcium sources had a significant effect on water use efficiency in BRS GA1 seedlings when irrigated with moderately saline water. The Ca-AA treatment stood out, showing an increase of 49.84% compared to the control treatment (Figure 1B). Calcium sources did not cause significant effects on WUE in BRS SC1 seedlings irrigated with moderately saline water.

The instantaneous carboxylation efficiency (iCE) did not vary between the passion fruit cultivars, regardless of irrigation with saline water or the application of Ca-complex sources, as shown in Figure 1C. In cv. BRS GA1, irrigation with moderately saline water resulted in a reduction in iCE in the treatments without calcium and with Ca-complexed with organic acids, but there was no significant difference between plants irrigated with low and moderate salinity water when applying Ca-complexed with amino acids (Figure 1C). On the other hand, in the BRS SC1 cultivar treated with Ca-AA, the iCE was reduced by 35% when irrigated with moderately saline water. In the W-Ca and Ca-OA treatments, there was no significant difference in iCE between the seedlings irrigated with moderate and low salinity water. The instantaneous carboxylation efficiency in the seedlings of the sour passion fruit cultivars did not differ significantly for the treatments without and with the application of complex calcium sources when irrigated with low salinity water (Figure 1C). However, in the BRS GA1 cultivar irrigated with moderately saline water, the iCE was higher in the Ca-AA treatments, being 43.71% higher than W-Ca, while in the BRS SC1 cultivar, there was no difference between the calcium sources.

These results indicate that the effect of moderately saline water varies between passion fruit cultivars and the calcium sources tested. In this condition, the seedlings of the BRS GA1 cultivar showed positive responses in photosynthetic rate and in water use and carboxylation efficiencies with the application of Ca-AA, while there were no differences in these variables in the seedlings of the BRS SC1 cultivar.

As for leaf transpiration, a reduction of 16.46% was observed in sour passion fruit seedlings of the BRS GA1 cultivar due to irrigation with saline water, as shown in Figure 2A. In contrast, no significant differences were found in the transpiration rate between the cultivars, regardless of the salinity of the irrigation water. These results highlight how the response of cultivars to saline water irrigation and the application of complex calcium can vary, affecting important variables such as carboxylation efficiency and leaf transpiration in different ways.

Leaf transpiration showed no significant differences between the sour passion fruit cultivars in the treatments with the different calcium sources (W-Ca, Ca-OA and Ca-AA), as illustrated in Figure 2B. In the BRS GA1 cultivar, it was observed that the plants that did not receive calcium showed higher leaf transpiration, while those treated with Ca complexed in organic acids showed lower leaf transpiration values. On the other hand, the application of calcium sources had no significant effect on the leaf transpiration of sour passion fruit seedlings cv. BRS SC1.

Irrigation with moderately saline water resulted in a reduction in the stomatal conductance of seedlings, regardless of the calcium source applied, with losses of 14.49, 22.03 and 9.96%, respectively, in the treatments without calcium, with calcium complexed in organic acids and with Ca-complexed in amino acids (Figure 2C). However, there were no significant differences in stomatal conductance between calcium sources in seedlings irrigated with saline water (low and moderate salinity).

Irrigation with moderately saline water increased internal CO_2_ concentration by 16.36% (Figure 3A), while reducing instantaneous water use efficiency by 29.67% (Figure 3B). It was observed that the application of Ca-AA resulted in a reduction in the CO_2_ concentration of seedlings, followed by the treatment with Ca-OA. On the other hand, the plants in the treatments without calcium had 18.80% higher internal CO_2_ concentrations than those treated with Ca-AA (Figure 3C).

### 2.2. Chlorophyll Indices and Chlorophyll a Fluorescence

The leaf chlorophyll indices of sour passion fruit seedlings responded to the interaction between cultivars, water salinity and sources of complex calcium, as shown in Table 2. The variables related to chlorophyll *a* fluorescence were not affected by the treatments.

In the seedlings irrigated with low salinity water, there was no difference in the chlorophyll *a* index (IChl*a*) between the cultivars in the treatment without calcium application (Figure 4A). Under irrigation with moderately saline water, sour passion fruit cv. BRS SC1 showed increases of 17.50% and 45.40% in the treatments without and with calcium complexed in organic acids, respectively. Irrigation with moderately saline water reduced the chlorophyll *a* indices of the cv. BRS GA1 in the treatment without calcium and with Ca-OA, while it increased the IChl*a* values when Ca-AA was applied (Figure 4A). For seedlings of the BRS SC1 cultivar, irrigation with moderately saline water increased IChl*a* by 29.60% in the treatments with calcium complexed in organic acids.

The seedlings of the cv. BRS SC1, irrigated with low salinity water and with the application of Ca-complexed with amino acids, showed a higher IChl*b* compared to the seedlings of cv. BRS GA1 (Figure 4B). Under conditions of irrigation with moderately saline water, the seedlings of cv. BRS SC1 treated with Ca-OA showed an IChl*b* value 114.96% higher than those from the other cultivar.

For the treatments without and with the application of Ca-OA, respectively, the seedlings of the cv. BRS GA1 showed a reduction in IChl*b* when irrigated with moderately saline water (Figure 4B). However, there was an increase in this variable with the application of Ca-AA, resulting in a 25.30% increase compared to the control treatment. The application of Ca-OA also increased the chlorophyll b index of BRS SC1 seedlings under moderately saline water conditions.

Under irrigation with moderately saline water, sour passion fruit seedlings cv. BRS SC1 showed higher IChl*t* compared to seedlings of cv. BRS GA1 in the treatments without calcium and with Ca-complexed with organic acids (Figure 4C). The total chlorophyll index of the cv. BRS GA1 was reduced in the treatments without calcium and with Ca-OA when irrigated with moderately saline water, with losses of 19.56% and 32.49%, respectively. The application of Ca-complexed with AA prompted an increment of 21.75% in the total IChl*t* index of BRS GA1 seedlings exposed to the moderately saline treatment. For seedlings of the BRS SC1 cultivar, there was no significant difference in the treatments without calcium and with Ca-AA at both salinity levels. However, in the treatments with Ca-OA, there was a 30.64% increase in IChl*t* when the seedlings were irrigated with moderately saline water.

The application of Ca-complexed with amino acids resulted in a higher IChl*t* in the cv. BRS GA1 irrigated with moderately saline water, with an increase of 18.7% compared to the treatment without calcium. There were no significant differences between calcium sources in the sour passion fruit cv. BRS SC1 when irrigated with moderately saline water.

Similar to Figure 1, the results indicate that the leaf chlorophyll indices in sour passion fruit seedlings irrigated with moderately saline water were higher with the application of calcium complexed in amino acids for the BRS GA1 cultivar, but there was no difference between the calcium sources in the BRS SC1 cultivar.

### 2.3. Electrolyte Leakage and Relative Water Content

The interaction cultivars × water salinity × complexed calcium sources had a significant effect on electrolyte leakage and relative water content in seedlings (Table 3).

Irrigation with moderately saline water increased electrolyte leakage, regardless of the application of complex calcium sources (Figure 5A). The highest EL values were observed in BRS SC1 seedlings irrigated with moderately saline water and Ca-OA (41.11%), while the lowest values were found in BRS GA1 seedlings irrigated with low salinity water and Ca-AA (12.98%).

Under the conditions of the substrate complexed without calcium, the BRS SC1 cultivar showed a lower value of electrolyte leakage compared to the BRS GA1 seedlings (Figure 5A). Under moderately saline water conditions, the application of complex calcium to BRS GA1 seedlings resulted in the lowest EL values (29.68%—Ca-OA and 28.61%—Ca-AA). In BRS SC1 seedlings, electrolyte leakage was reduced with the application of Ca-AA (26.29%), although it did not differ statistically from the treatment without calcium (29.19%).

Figure 5B shows that, regardless of the calcium sources, there was no significant difference in relative water content (RWC) between the passion fruit cultivars irrigated with low salinity water. However, under irrigation with moderately saline water, it was observed that in the treatments without the application of calcium sources, the seedlings of the cv. BRS GA1 (95.74%) had a higher RWC compared to cultivar BRS SC1 (82.84%).

Irrigation with moderately saline water increased the relative water content in most treatments, especially in the treatments with calcium complexed in amino acids in both cultivars. There was an increase from 81.29% to 87.44% in cv. BRS GA1 and from 79.61% to 91.40% in cv. BRS SC1. Under low salinity conditions, no statistical differences were observed between the calcium sources.

However, the cultivars behaved differently when irrigated with moderately saline water. In the BRS GA1 seedlings, the highest RWC values were found in the treatments without calcium application, with an increase of 8.98% over the Ca-AA treatments. On the other hand, in the seedlings of the BRS SC1 cultivar, the highest RWC values were observed in the treatments with Ca-AA (91.40%), representing an increase of 10.32% over the treatment without calcium.

In summary, the results indicate that electrolyte leakage from seedlings irrigated with moderately saline water increased in both cultivars when Ca-AA was applied as a stress attenuator.

### 2.4. Multivariate Analysis of Principal Components

Based on the principal component analysis (PCA) of the physiological traits of the sour passion fruit cv. BRS GA1 (Figure 6), it was observed that the PC1 and PC2 vectors explained 52.8% and 19% of the variance, respectively, totaling 71.8% of the total variance of the data.

Irrigation with moderately saline water had a significant impact on the variables of relative water content, electrolyte leakage and internal CO_2_ concentration of sour passion fruit cv. BRS GA1, especially in the treatments without and with Ca-complexed with organic acids. In contrast, seedlings irrigated with low salinity water showed increases in gas exchange variables (stomatal conductance—gs, transpiration rate—E, CO_2_ assimilation rate—A, water use efficiency—EUA, instantaneous water use efficiency—iWUE and instantaneous carboxylation efficiency—iCE), and chlorophyll indices (IChl*a*, IChl*b* and IChl*t*), regardless of the application of calcium sources.

When analyzing the principal components of the physiological variables of the BRS SC1 seedlings (Figure 7), it was observed that the eigenvalues accounted for 37.6% (PC1) and 21% (PC2) of the variance, totaling 58.6% of the total variance of the data.

This analysis indicates that there was a significant distribution between the main components, reflecting the diversity in the physiological responses of the plants to the different treatments applied, such as irrigation with low and moderate salinity water and the application of complex calcium.

Irrigation with low salinity water associated with the application of Ca-complexed with amino acids promoted higher values of CO_2_ assimilation rate (A), water use efficiency (WUE), intrinsic water use efficiency (iWUE) and instantaneous carboxylation efficiency (iCE) in the seedlings of the cv. BRS SC1. These variables showed a negative correlation with internal CO_2_ concentration (Ci), transpiration rate (E) and relative leaf water content (RWC), which responded more strongly to irrigation with moderately saline water and the application of complex calcium sources.

Initial fluorescence (F_0_), stomatal conductance (gs) and leaf transpiration (E) were most influenced by irrigation with low salinity water associated with the application of calcium complexed with organic acids (Ca-OA). On the other hand, irrigation with moderately saline water had a high influence on the maximum chlorophyll fluorescence (F_m_), variable fluorescence (F_v_) and quantum efficiency of photosystem II (RQPII) of sour passion fruit cv. BRS SC1. The seedlings’ leaf chlorophyll indices were little influenced by the application of the treatments.

## 3. Materials and Methods

### 3.1. Description of the Experiment Site

The experiment was conducted between October and December 2023, in a protected environment (green house) located in the Department of Agronomic and Forestry Sciences (DAFS) belonging to the Federal Rural University of the Semi-Arid (UFERSA), which is located in the municipality of Mossoró, Rio Grande do Norte, Brazil. The protected environment is georeferenced by the coordinates 5°12′12″ S and 37°19′26″ W at an altitude of 17 m. It is characterized by transparent low-density polyethylene cover, treated against the action of ultraviolet rays, in the shape of an arch, with dimensions of 6.5 m × 12 m. The sides and front walls were made with anti-aphid screens (50%) and a 0.24 m masonry plinth.

Sour passion fruit seedlings were installed on benches. During the assay, the average daily temperatures and relative humidity of air inside the protected environment were recorded using an HTC-1 Clock/Humidity Digital LCD temperature and humidity meter (Beijing, China), as presented in Figure 8.

### 3.2. Experimental Design and Treatments

The experiment was carried out in a randomized block design, in a 2 × 2 × 3 factorial scheme, with three replications and two plants per plot. The treatments referred to two cultivars of sour passion fruit (BRS Gigante Amarelo—BRS GA1 and BRS Sol do Cerrado—BRS SC1), irrigated with low salinity and moderately saline water (electrical conductivity of 0.5 and 4.0 dS m^−1^, respectively) and the application of three sources of Ca complexed as a water salinity attenuator (control—without calcium [W-Ca], calcium complexed in organic acids [Ca-OA] and calcium complexed in amino acids [Ca-AA]).

The F1 seeds of the passion fruit cultivars BRS GA1 and BRS SC1 were obtained from mother plants in Embrapa Cerrado, Brasilia, Brazil. The low salinity water (0.5 dS m^−1^) came from the local municipal water supply, while the moderately saline water (4.0 dS m^−1^) was obtained by diluting the stock solution (8.0 dS m^−1^), prepared with P.A. sodium chloride (NaCl; PM = 58.44), hydrated calcium chloride (CaCl_2_·2H_2_O; PM = 147.01) and magnesium chloride hexahydrate (MgCl_2_·6H_2_O; PM = 203.30) in an equivalent ratio of 7:2:1, in supply water (0.5 dS m^−1^), following the relationship between ECw and its concentration (mmol_c_ L^−1^ ≈ EC × 10), according to the methodology contained in Silva [21]. This cation ratio (7:2:1) represents the chemical composition of most surface and underground saline water reservoirs in the Brazilian northeast. The calcium sources used were the commercial products Codasal^®^ (Sustainable Agro Solution, Almacelles, Spain) and Hendosar^®^ (Adriatica S.p.A., Loreo, Italy), respectively, as a source of Ca-complexed in organic acids and amino acids at a concentration of 2% (volume basis).

### 3.3. Conducting the Experiment

The substrate used to grow the seedlings was made up of a mixture of soil collected from a depth of 0–0.20 m and well-decomposed bovine manure in a ratio of 2:1 (*v/v*), the chemical composition of which was analyzed according to the methodologies presented by Silva [22]. The results of the analysis are shown in Table 4.

Subsequently, the substrate (1.7 kg) was placed in black polyethylene bags with a capacity of 1.5 dm^3^; three seeds were placed in each bag at a depth of 1 cm, after which they were covered with a thin layer of substrate. Emergence began at 7 days after sowing (DAS) and stabilized at 28 DAS, counting as normal seedlings those that had fully emerged from the substrate and had fully expanded cotyledons. Ten days after emergence had stabilized, the seedlings were thinned out, leaving the most vigorous one per polyethylene bag.

The plants were irrigated daily with the respective irrigation water using the weighing method, replenishing the volume of water corresponding to the evapotranspiration of the plants of the previous day (24 h), so as to leave the moisture content of the substrate at 90% of its field capacity [23]. To do this, before setting up the experiment, ten polyethylene bags containing substrate were selected at random, weighed and then volumes of water were added gradually until all the pore spaces in the substrate were filled and the water began to drain. Immediately, the volume of water drained was measured in a measuring cylinder and the polyethylene bags were weighed to determine the weight of the bags at field capacity. The amount of water to be applied corresponds to the difference between the weight of the polyethylene bag at 90% field capacity and the weight on the day of irrigation. When the weight of the bag with the substrate was the same as that at 90% field capacity, the sour passion fruit seedlings were not irrigated.

The attenuators (Codasal^®^ and Hendosar^®^) were applied at 12 and 32 days after emergence in a volume equivalent to 50 mL per plant (solution diluted to 2% in water). The solution was prepared by diluting the attenuators in low salinity water (0.5 dS m^−1^), according to the manufacturer’s recommendations. The solution was applied gradually to the surface of the substrate until maximum infiltration was achieved. Codasal^®^ is a dark-colored liquid organo-mineral fertilizer that is Ca-complexed with organic acids (lignosulphonate) and contains a 6.0% N, 8.7% CaO and 14.7% lignosulphonate complexing agent and a salt index of 40.74%. Hendosar^®^ is a light-red liquid mineral fertilizer containing Ca-complexed with amino acids, with 9.0% N, 6.0% K_2_O, 7.5% Ca, 1.2% Mg and a salt index of 49.52%.

### 3.4. Traits Analysed

#### 3.4.1. Chlorophyll a Fluorescence

At 60 DAS, the chlorophyll a fluorescence of the leaves was measured using the following variables: initial fluorescence (F_0_), maximum fluorescence (F_m_), variable fluorescence (F_v_) and photosystem II quantum yield (RQPII) using a PEAITM pulse-modulated Plant Efficiency Analyzer (Hansatech Instruments Co., King’s Lynn, UK). For this, three plants per treatment were selected for evaluation, in which the third pair of fully expanded leaves was selected, counting from the apex to the base of the plants. Measurements were taken between 08:00 and 11:00 a.m. after the leaves were adapted to the dark for at least 30 min.

#### 3.4.2. Leaf Chlorophyll Indices

The leaf chlorophyll indices of the sour passion fruit seedlings were measured on the same leaves selected for chlorophyll *a* fluorescence analysis at 60 DAS. The chlorophyll *a* index (IChl*a*), chlorophyll *b* index (IChl*b*) and chlorophyll total index (IChl*t*) were measured using a Falker^®^ ChlorofiLOG portable meter (Porto Alegre, Brazil). On each plant, three readings were taken on the basal, median and apical part of the leaf and the average was calculated from these data.

#### 3.4.3. Gas Exchange

At 60 DAS, the third or fourth pair of leaves on the seedlings was selected to analyze gas exchange using the following variables: stomatal conductance (gs—mol H_2_O m^−2^ s^−1^); net photosynthesis rate (A—µmol CO_2_ m^−2^ s^−1^); internal CO_2_ concentration (Ci—µmol CO_2_ m^−2^ s^−1^); and leaf transpiration (E—mmol H_2_O m^−2^ s^−1^), measured with a portable infrared carbon dioxide analyzer (IRGA), model LCorp-SD from BioScientific^®^ (Hoddesdon, UK), adjusted to an air temperature of 25 °C, irradiance of 1. 200 µmol m^−2^ s^−1^ and air flow of 200 mL min^−1^ [23]. The readings were taken in the morning, between 08:00 and 11:00 a.m. From these data, water use efficiency (WUE) was determined by the A/E ratio and intrinsic water use efficiency (iWUE) by the A/gs ratio.

The relative water content (RWC) was determined according to Beltrano and Ronco [24], adapted. Leaf disks of 0.7 cm in diameter were removed from the central part of the leaf vein located in the middle leaves of the seedlings to determine the RWC. After determining the mass of fresh matter (MFM), the disks were placed in distilled water for 4 h. The turgid samples were quickly dried with paper towel to determine the turgid weight (WT). The disks were then placed in an oven at 80 °C for 24 h to determine the dry matter weight (DMW). The following equation was used for this: RWC (%) = (MFM − DMW)/(WT − DMW) × 100.

#### 3.4.4. Electrolyte Leakage

Electrolyte leakage (EL) was determined by removing leaf disks with diameters of 0.7 cm, according to the methodology proposed by Beltrano and Ronco [24], with small adaptations. After being washed three times in distilled water for 3 min, 10 leaf disks were placed in 25 mL of deionized water in Falcon tubes, and the initial EC of the contents was read after 90 min. Once this was carried out, the Falcon tubes were taken to a water bath at 85 °C for 4 h, then cooled to a temperature of 25 °C; the final EC was read and the EL was calculated using the following formula: EL = (L1/L2) × 100, where L1—initial EC reading; L2—final EC reading.

### 3.5. Statistical Analysis

Before the statistical analysis, the data were evaluated for normality and homogeneity of variance using the Shapiro–Wilk test. Subsequently, the analysis of variance was carried out using the F test (*p* ≤ 0.05). The means for the cultivars, electrical conductivities of the water and calcium sources were compared using the Tukey test at 0.05 probability. The statistical software Sisvar 5.6 [25] was used to analyze the data. Figures were drawn using Sigmaplot software version 12.5. Multivariate analysis was carried out using principal component analysis (PCA) with the statistical analysis packages available in R Studio version 4.4.0 [26].

## 4. Discussion

Irrigation with moderately saline water resulted in a reduction in photosynthetic rate, water use efficiency and instantaneous carboxylation efficiency in seedlings (Figure 1). Photosynthesis is the crucial physicochemical process in plant metabolism and is highly sensitive to environmental changes under salt stress; its efficiency is often reduced, either by stomatal and/or non-stomatal limitations [7]. However, the photochemical efficiency of PSII was not affected by irrigation with moderately saline water; the seedlings’ temporary stress may have adjusted their photo protection mechanisms (physiological plasticity) without significantly compromising the photochemical efficiency of the plants [27].

The concentrations of sodium and chloride salts in water can significantly alter the physiological and biochemical functions of plants, restricting the growth and development of crops [28], including during the vegetative phase of the sour passion fruit [9], a fact observed in this study during the initial growth of the plants. Exposure to salinity can reduce stomatal conductance due to the partial closure of stomata and decreases in the influx of carbon dioxide and, consequently, the photosynthetic rate, which is limited by the reduced availability of CO_2_ for the Calvin–Benson cycle [5,28].

The application of Ca-complexed with amino acids reduced the damage caused by toxic salts in the photosynthetic activity and efficiency of seedlings (Figure 1). In addition, studies have shown that the application of calcium in the form of calcium pyruvate also attenuated the adverse effects of salinity on Redondo Amarelo passion fruit seedlings [10]. These effects are attributed to calcium, which plays a crucial role in maintaining the structural and functional integrity of cell membranes and the cell wall, as well as regulating ion transport and the enzymatic activity of the cell wall [9,10,13]. In addition, the increase in Ca^2+^ in the substrate improves K^+^/Na^+^ selectivity, altering the absorption rate in favor of K^+^ to the detriment of Na^+^, reducing Na^+^ toxicity in the plant [13,29].

The application of amino acids as a complexing agent may also improve the process of plant acclimatization to abiotic stress conditions, such as salinity, by acting as osmolytes, facilitating the transport of ions, promoting protein synthesis and maintaining the integrity of biomembranes [17]. Furthermore, the source of Ca-complexed with amino acids had in its composition higher percentages of nitrogen, potassium and magnesium, which are involved in the production of proline (osmoregulator and osmoprotector), stomatal regulation, photosynthetic efficiency, protein synthesis and maintenance of the water status of plants, reducing the effect of saline stress [30]. Concerning studies with broccoli var. Italica, Haghighi et al. [31] showed that foliar application of calcium complexed with tryptophan increased the photosynthetic rate of plants under salt stress, highlighting the potential of calcium–amino acid complexes in mitigating the adverse effects of salinity on the crop.

Irrigation with moderately saline water reduced the leaf transpiration of sour passion fruit cv. BRS GA1 (Figure 2A), due in part to the partial closure of the stomata [25,28,32], as seen in Figure 2C. This effect occurs due to osmotic stress, which inhibits the absorption of water by plants, limiting the flow of water vapor into the atmosphere, as described by Lima et al. [6] in studies with seedlings of three cultivars of sour passion fruit irrigated with moderately saline water (3.5 dS m^−1^). Nevertheless, Paiva et al. [33] found that there is variability in physiological responses among sour passion fruit cultivars (BRS GA1, BRS S1 and SCS 437 Catarina) irrigated with moderately saline water (3.5 dS m^−1^) at 154 days after transplanting, in which the BRS GA1 cultivar showed higher transpiration and stomatal conductance rates compared to the other cultivars.

Calcium lignosulfonate, calcium complexed in organic acids, as mentioned by Thye et al. [19], disassociates into Ca^2+^ ions and lignosulfonate, thus improving the bioavailability of calcium to plants. These calcium ions can be absorbed and transported by plant cells via calcium ion channels. Under conditions of water restriction due to the high salinity of the substrate, this calcium may contribute to stomatal closure, reducing seedlings’ transpiration rate (Figure 2B).

The increase in internal CO_2_ concentration seen in seedlings irrigated with moderately saline water (Figure 3A) might be due to the progressive accumulation of salts in the photosynthetically active mesophyll tissues, which leads to an inhibition of CO_2_ assimilation, mainly affecting the photosynthetic processes in the chloroplast, also known as non-stomatal limitations [20], as well as due to stomatal constraints (Figure 2C) that may limit CO_2_ efflux. These limitations involve suppression of photosynthetic enzymes of the Calvin cycle, disruption of chlorophyll biosynthesis, reduced operational efficiency and structural integrity of the photosynthetic apparatus and the membranes of the thylakoids [33,34].

The seedlings reduced their intrinsic water use efficiency (iWUE) when irrigated with moderately saline water (Figure 3B). Intrinsic water use efficiency refers to the amount of carbon dioxide fixed by the plant for each unit of H_2_O that is lost in the fixation process and under conditions of water–salt stress. Under this situation, plants reduce their absorption of water from the soil and, as a way of avoiding loss to the external environment, they acclimatize by reducing the opening of the sub-stomatal chambers and restricting the influx of CO_2_ [35]. Similar behavior was observed by Lima et al. [36] in sour passion fruit cv. BRS GA1 under different irrigation strategies with saline water and potassium fertilization, when they found that regardless of the phenological stage, iWUE was reduced when the plants were irrigated with moderately saline water.

The use of Ca-complex reduced the internal concentration of CO_2_ in the cells of seedlings, especially with Ca-complexed in amino acids (Figure 3C). The application of complexing or chelating agents protects the calcium applied to the substrate from various harmful reactions, such as fixation, precipitate formation and leaching, resulting in greater efficiency of the attenuator and a better rate of absorption and use by the plants [17]. In addition, amino acids can promote the development of faster and more effective nutrient absorption channels or mechanisms in root cells, optimizing the transport and translocation of calcium by plant cell membrane transporters [18].

Adequate levels of Ca^2+^ ions in the chloroplast regulate the photosynthetic pathway [37]. In light-dependent reactions, calcium acts as a cofactor in the formation of the active site in photosystem II and facilitates the oxidation of water to form ATP [38]. In addition, calcium increases CO_2_ consumption by regulating enzymes involved in the Calvin cycle, increasing the plant’s photosynthetic capacity and organic compound formation [19].

In general, the leaf chlorophyll indices of seedlings were reduced by irrigation with moderately saline water (Figure 4). High levels of salinity are known to affect plant photosynthesis through non-stomatal limitations, including changes in chlorophyll concentration. This effect can be attributed to an increase in the activity of chlorophyllase, an enzyme responsible for degrading chlorophyll [8].

In Figure 4, it is observed that the application of complexed calcium (organic acids and amino acids) reduced the effects of salinity on chlorophyll indices. Similarly, Bezerra et al. [39] observed increases in leaf chlorophyll indices in adult passion fruit plants irrigated with calcium doses. The authors point out that although calcium is not part of the chlorophyll molecule, it has a synergistic effect with nitrogen, making it possible to increase chlorophyll synthesis. In addition to calcium, the complexing agents in the form of organic acids or amino acids present in these attenuating agents generally contain nitrogen in various forms. These compounds contribute to osmoregulation, chlorophyll biosynthesis, and leaf expansion [17]. Despite not responding to the treatments (Table 3), the F_0_, F_v_, F_m_ and RQPII variables showed mean values of 72.25, 262.72, 334.42 and 0.77, respectively. Similar results were observed by Bezerra et al. [40], who found no influence of water salinity and calcium application on chlorophyll fluorescence variables. Values for the maximum photochemical yield of PSII ranging from 0.80 to 0.86 indicate that the plants are not under stress [41]. Therefore, as the values obtained in our research were below this range (0.77), it is possible that the stress was not strong enough to significantly alter PSII photochemistry.

Irrigation with moderately saline water causes cell damage in seedlings, increasing intracellular electrolyte leakage (Figure 5A). Plasma membranes are the primary sites of damage from salinity-specific ions (Na^+^ and Cl^−^) [42,43]. According to the same authors, maintaining cell membrane stability under conditions of salt stress is one of the most important selection criteria for identifying salt-tolerant species or cultivars. The sour passion fruit cultivars showed greater cell membrane stability with the application of complexed calcium sources (Figure 5A). The calcium chelates (Ca-OA or Ca-AA) present in these sources can influence the distribution of Na and Cl, reducing the phytotoxic effects on plant metabolism, significantly protecting against lipid peroxidation and increasing the integrity of cell membranes under salt stress [17]. In addition, Ca^2+^ is a particularly important nutrient for plants exposed to salt stress (NaCl), due to its role in reducing Na^+^ absorption and increasing the absorption of K^+^ and Ca^2+^ itself, which contributes to membrane stability and plant growth [13].

A higher RWC was observed in the sour passion fruit seedlings of cv. BRS GA1 compared to cv. BRS SC1 (Figure 5B). Normally, the presence of salts in the root zone can trigger osmotic stress, hindering the plant’s ability to extract water from the soil, reducing the conductivity of the water in the root and, consequently, the relative water content in the cells [8]. Thus, maintaining a high RWC under conditions of salt stress can be a criterion for identifying and selecting materials that are more tolerant to salinity [6]. Sour passion fruit cultivars that have the ability to adjust more effectively to the unfavorable conditions of low water availability in the substrate are considered to be more tolerant to salinity.

The sources of complex calcium increased the ability of seedlings to maintain high levels of cellular RWC (Figure 5B). The addition of calcium-complexing agents (such as organic acids and amino acids) as Ca^2+^ stabilizers promotes the absorption of other organic and inorganic compounds (betaine, glycine, acids, calcium and potassium ions) by the plant, facilitating osmotic adjustment and strengthening the cell wall, which provides greater plant resistance to water stress [17]. Similar behavior was observed in barley cultivars (*Hordeum vulgare* L.) under salt stress (NaCl levels up to 300 mM), where the application of calcium complexed with lignosulfonate helped to mitigate the effects of salinity in water with NaCl levels of 200 mM [19].

## 5. Conclusions

The use of Ca-complexed sources has the potential to mitigate the deleterious effects of irrigation with moderately saline water (electrical conductivity of 4.0 dS m^−1^) on the physiological parameters of the seedlings of sour passion fruit cultivars at 60 days after sowing. The cultivar BRS SC1 proved to be more tolerant to irrigation with moderately saline water during seedling formation. This includes the promotion of better photosynthetic efficiency, higher chlorophyll indices and cell membrane stability, demonstrated by lower leakage and higher relative water content. Based on these results, it is recommended that future research focus on investigating the role of complex Ca^2+^ sources in modulating the biochemical activity and enzymatic defenses of sour passion fruit under conditions of moderate to severe salinity, as a strategy to increase salt tolerance and enable the sustainable insertion of agricultural areas marginalized by salinity problems.

## Figures and Tables

**Figure 1 plants-13-02912-f001:**
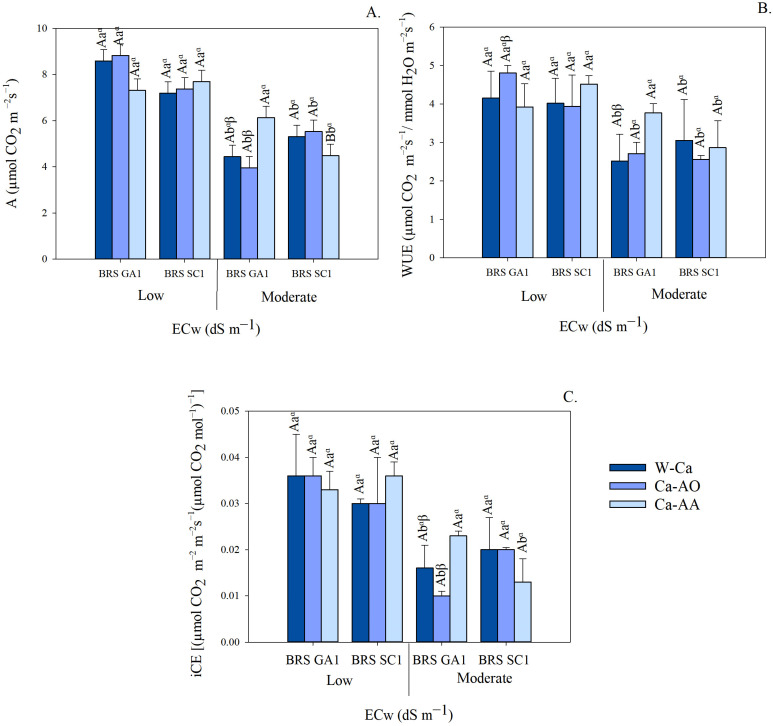
Net assimilation rate—A (**A**), water use efficiency—WUE (**B**) and instantaneous carboxylation efficiency—iCE (**C**) of seedlings of sour passion fruit cultivars irrigated with low and moderate salinity water under application of Ca-complexed sources as attenuators. Means followed by same uppercase letters do not differ for cultivars within each irrigation water salinity and application of calcium sources, according to F test (*p* ≤ 0.05). Means followed by same lowercase letters do not differ for irrigation water salinity within each cultivar and calcium sources, by F test (*p* ≤ 0.05). Means followed by same Greek letters do not differ for calcium sources within each cultivar and irrigation water salinity, using Tukey test (*p* ≤ 0.05). Scatter above bar represents average standard deviation (*n* = 3).

**Figure 2 plants-13-02912-f002:**
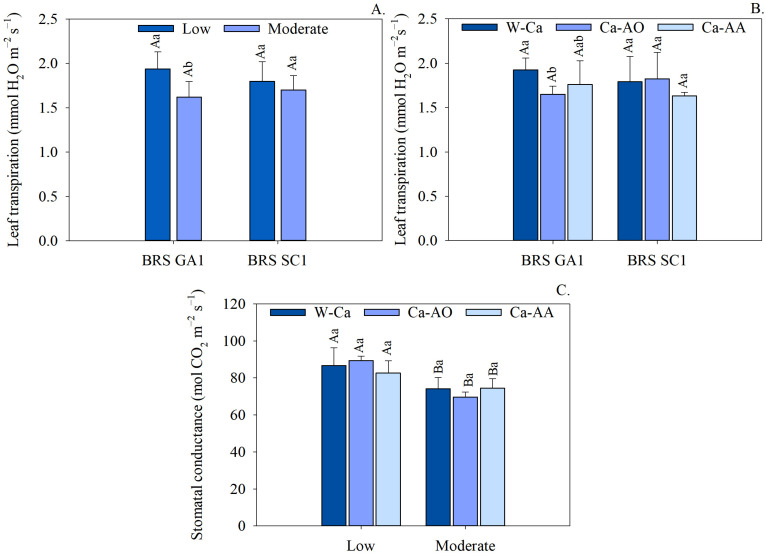
Leaf transpiration in seedlings of sour passion fruit cultivars irrigated with low and moderate salinity water (**A**), application of complex calcium sources (**B**) and internal CO_2_ concentration of sour passion fruit seedlings irrigated with low and moderate salinity water and application of complex calcium sources (**C**). (**A**) Means followed by same uppercase letters do not differ for cultivars within each irrigation water salinity by F test (*p* ≤ 0.05) and means followed by same lowercase letters do not differ for irrigation water salinity within each cultivar by F test (*p* ≤ 0.05). (**B**) Means followed by same uppercase letters do not differ for cultivars within each calcium source by F test (*p* ≤ 0.05) and means followed by same lowercase letters do not differ for calcium sources within each passion fruit cultivar by Tukey test (*p* ≤ 0.05). (**C**) Means followed by same uppercase letters do not differ from each other for salinity of irrigation water within each calcium source by F test (*p* ≤ 0.05) and means followed by same lowercase letters do not differ from each other for calcium sources within each salinity of irrigation water by Tukey test (*p* ≤ 0.05). Scatter above bar represents average standard deviation (*n* = 3).

**Figure 3 plants-13-02912-f003:**
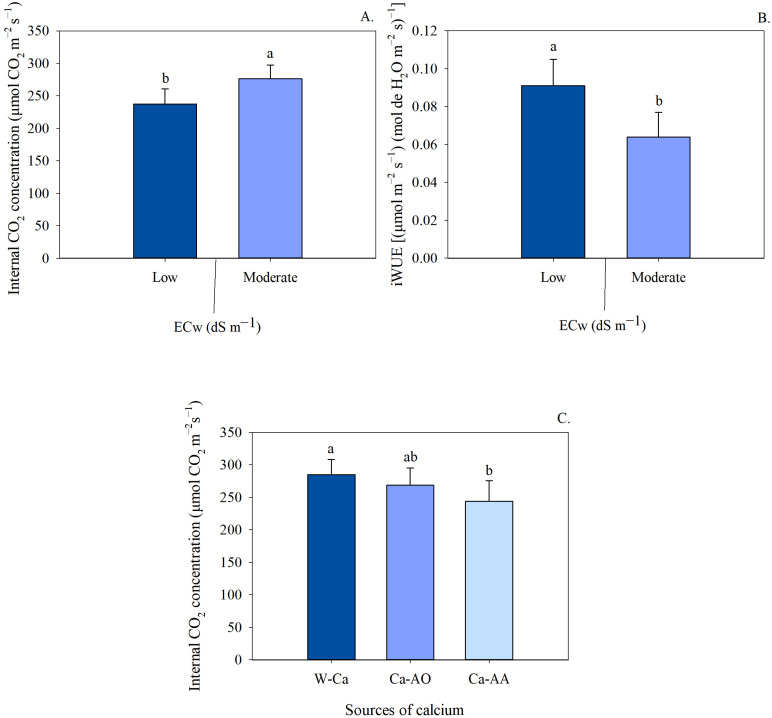
Leaf transpiration in seedlings of sour passion fruit cultivars irrigated with low and moderate salinity water (**A**), application of complex calcium sources (**B**) and stomatal conductance of sour passion fruit seedlings irrigated with low and moderate salinity water and application of complex calcium sources (**C**). (**A**,**B**) Means followed by same lowercase letters do not differ for irrigation water salinity according to F test (*p* ≤ 0.05). (**C**) Means followed by same lowercase letters do not differ for calcium sources using Tukey test (*p* ≤ 0.05). Scatter above bar represents average standard deviation (*n* = 3).

**Figure 4 plants-13-02912-f004:**
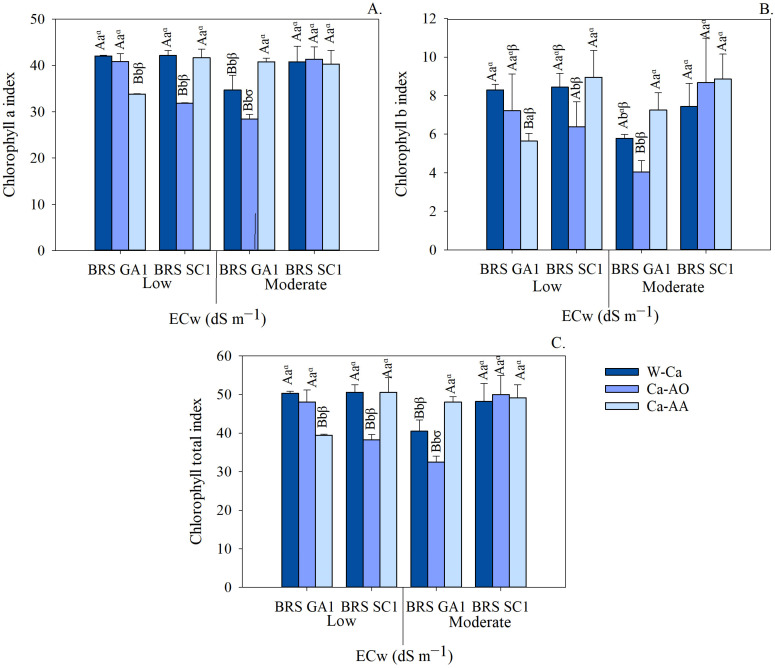
Chlorophyll *a* (**A**), chlorophyll *b* (**B**) and total chlorophyll (**C**) indices of seedlings of sour passion fruit cultivars irrigated with low and moderate salinity water under application of Ca-complex sources as attenuators. Means followed by same uppercase letters do not differ for cultivars within each irrigation water salinity and application of calcium sources, according to F test (*p* ≤ 0.05). Means followed by same lowercase letters do not differ for irrigation water salinity within each cultivar and calcium sources, by F test (*p* ≤ 0.05). Means followed by same Greek letters do not differ for calcium sources within each cultivar and irrigation water salinity, using Tukey test (*p* ≤ 0.05). Scatter above bar represents average standard deviation (*n* = 3).

**Figure 5 plants-13-02912-f005:**
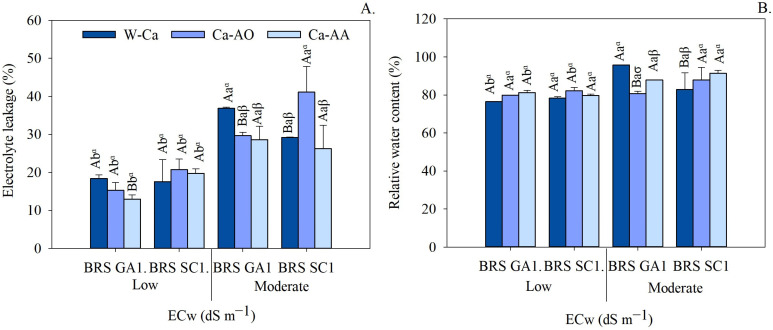
Electrolyte leakage (**A**) and relative water content (**B**) of seedlings of sour passion fruit cultivars irrigated with low and moderate salinity water under application of calcium complex sources as attenuators. Means followed by same uppercase letters do not differ for cultivars within each irrigation water salinity and application of calcium sources by F test (*p* ≤ 0.05). Means followed by same lowercase letters do not differ for irrigation water salinity within each cultivar and calcium sources by F test (*p* ≤ 0.05). Means followed by same Greek letters do not differ for calcium sources within each cultivar and irrigation water salinity by Tukey test (*p* ≤ 0.05). Scatter above bar represents mean standard deviation (*n* = 3).

**Figure 6 plants-13-02912-f006:**
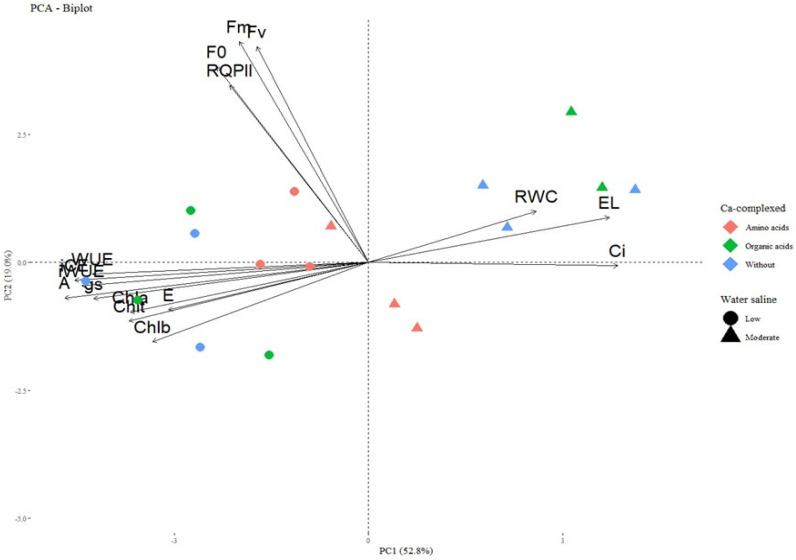
Principal component analysis (PCA) for the physiological traits of sour passion fruit seedlings cv. BRS GA1 irrigated with low and moderate salinity water under sources of complexed calcium as a mitigator (*n* = 3).

**Figure 7 plants-13-02912-f007:**
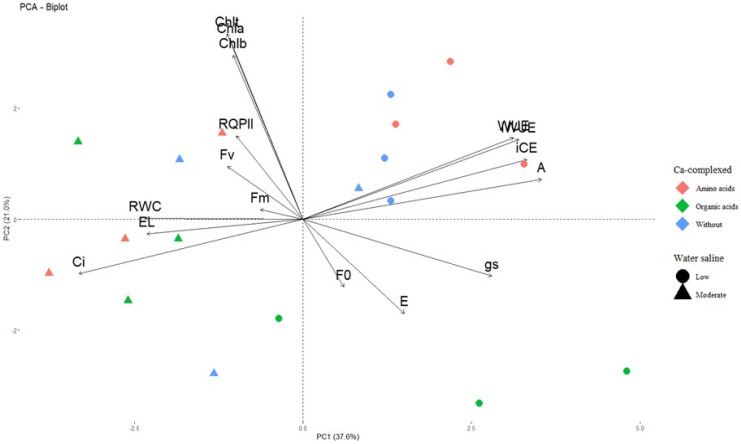
Principal component analysis (PCA) for the physiological traits of sour passion fruit seedlings cv. BRS SC1 irrigated with low and moderate salinity water under sources of calcium complexed as an attenuator (*n* = 3).

**Figure 8 plants-13-02912-f008:**
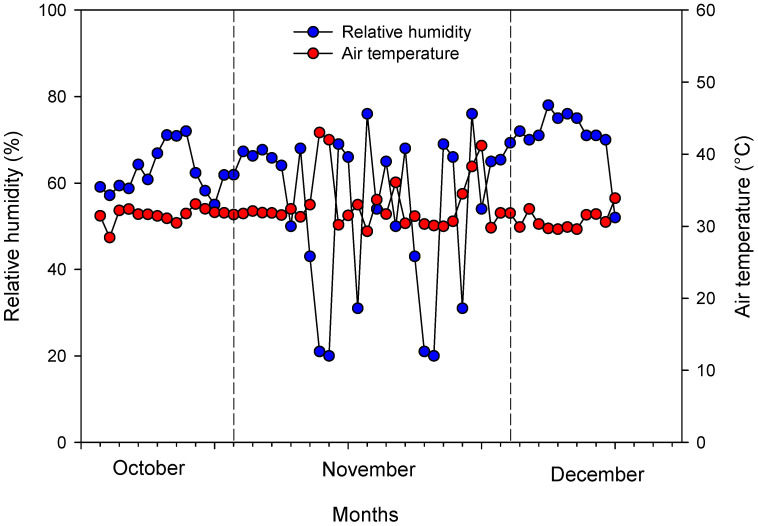
Temperature and relative humidity of air inside the protected environment during the experimental period (18 October to 18 December 2024).

**Table 1 plants-13-02912-t001:** Summary of analysis of variance for gas exchange variables in sour passion fruit cultivars (Cul) irrigated with saline water (Sal), and application of complex calcium sources (Cal) as salinity attenuators at 60 days after sowing.

S.V.	D.F.	Mean Squares			
		E	gs	A	
Cul	2	0.007 ^ns^	32.832 ^ns^	1.782 ^ns^	
Sal	1	0.390 **	1638.610 **	82.344 **	
Cal	2	0.084 *	10.737 ^ns^	0.196 ^ns^	
Cul × Sal	1	0.110 *	38.274 ^ns^	1.288 ^ns^	
Cul × Cal	2	0.092 *	69.151 ^ns^	0.104 ^ns^	
Sal × Cal	2	0.002 ^ns^	100.330 *	2.079 ^ns^	
Cul × Sal × Cal	2	0.005 ^ns^	95.102 ^ns^	4.367 *	
Block	2	0.138	19. 543	1.369	
Residue	22	0.023	28.303	0.913	
Total	35	-	-	-	
CV (%)	-	8.61	4.83	6.69	
		Ci	WUE	iWUE	iCE
Cul	2	15.853 ^ns^	0.213 ^ns^	0.001 ^ns^	0.001 ^ns^
Sal	1	13,599.058 *	15.510 **	0.006 **	0.002 ^ns^
Cal	2	1854.504 *	0.374 ^ns^	0.001 ^ns^	0.001 ^ns^
Cul × Sal	1	108.889 ^ns^	0.002 ^ns^	0.001 ^ns^	0.001 ^ns^
Cul × Cal	2	413.560 ^ns^	0.381 ^ns^	0.001 ^ns^	0.001 ^ns^
Sal × Cal	2	290.317 ^ns^	0.514 ^ns^	0.001 ^ns^	0.001 ^ns^
Cul × Sal × Cal	2	786.346 ^ns^	1.203 *	0.003 ^ns^	0.001 *
Block	2	1528.473	0.992	0.002	0.001
Residue	22	243.847	0.318	0.001	0.000
Total	35		-	-	-
CV (%)	-	6.08	15.81	14.40	24.00

SV = source of variation; DF = degree of freedom; CV = coefficient of variation; ^ns^, * and ** = not significant, significant at 0.05 and 0.01 probability, respectively. E—leaf transpiration; gs—stomatal conductance; A—net photosynthesis rate; Ci—internal CO_2_ concentration, WUE—water use efficiency; iWUE—intrinsic water use efficiency; iCE—instantaneous carboxylation efficiency.

**Table 2 plants-13-02912-t002:** Summary of analysis of variance, referring to the variables leaf chlorophyll index and chlorophyll fluorescence in the cultivars of sour passion fruit (Cul) irrigated with saline water (Sal) and application of Ca-complexed sources (Cal) as salinity attenuators at 60 days after sowing.

S.V.	D.F.	Mean Squares			
		IChl*a*	IChl*b*	IChl*t*	
Cul	2	76.038 **	27.825 **	195.860 **	
Sal	1	9.261 ^ns^	2.068 ^ns^	20.085 ^ns^	
Cal	2	62.779 **	4.145 ^ns^	97.511 **	
Cul × Sal	1	94.997 **	7.084 *	153.966 **	
Cul × Cal	2	2.203 ^ns^	1.815 ^ns^	4.704 ^ns^	
Sal × Cal	2	39.151 **	4.689 *	70.693 **	
Cul × Sal × Cal	2	171.36 **	9.689 *	262.611 **	
Block	2	14.086	4.492	34.096	
Residue	22	3.393	1.323	6.376	
Total	35	-	-	-	
CV (%)	-	4.82	15.86	5.55	
		F_0_	F_m_	F_V_	RQPII
Cul	2	380.250 ^ns^	1980.250 ^ns^	245.444 ^ns^	0.002 ^ns^
Sal	1	406.694 ^ns^	4074.694 ^ns^	1089.000 ^ns^	0.001 ^ns^
Cal	2	88.583 ^ns^	238.083 ^ns^	149.527 ^ns^	0.009 ^ns^
Cul × Sal	1	72.250 ^ns^	793.361 ^ns^	81.000 ^ns^	0.002 ^ns^
Cul × Cal	2	424.083 ^ns^	1629.250 ^ns^	440.527 ^ns^	0.001 ^ns^
Sal × Cal	2	105.194 ^ns^	189.194 ^ns^	207.750 ^ns^	0.002 ^ns^
Cul × Sal × Cal	2	217.583 ^ns^	12,097.694 ^ns^	1170.083 ^ns^	0.010 ^ns^
Block	2	991.083	1160.583	138.861	0.001
Residue	22	179.659	2204.765	1720.830	0.005
Total	35	-	-	-	-
CV (%)	-	18.30	14.04	15.79	9.95

SV = source of variation; DF = degree of freedom; CV = coefficient of variation; ^ns^, * and ** = not significant, significant at 0.05 and 0.01 probability, respectively. IChl*a*—chlorophyll *a* index; IChl*b*—chlorophyll *b* index; ICh*t*—chlorophyll total index; F_0_—initial fluorescence; F_m_—maximum fluorescence; F_v_—variable fluorescence; RQPII—quantum yield of photosystem II.

**Table 3 plants-13-02912-t003:** Summary of analysis of variance for electrolyte leakage and relative water content in sour passion fruit cultivars (Cul) irrigated with saline water (Sal) and application of Ca-complex sources (Cal) as salinity attenuators at 60 days after sowing.

S.V.	D.F.	Mean Squares	
		EL	RWC
Cul	2	41.667 ^ns^	0.043 ^ns^
Sal	1	18.94.135 **	588.790 **
Cal	2	74.488 *	17.423
Cul × Sal	1	25.050 ^ns^	5.405
Cul × Cal	2	119.695 **	81.170 *
Sal × Cal	2	30.544 ^ns^	57.721 **
Cul × Sal × Cal	2	49.778 *	98.228 **
Block	2	24.809	22.508
Residue	22	11.285	10.002
Total	35	-	-
CV	-	13.60	3.78

SV = source of variation; DF = degree of freedom; CV = coefficient of variation; ^ns^, * and ** = not significant, significant at 0.05 and 0.01 probability, respectively. EL—electrolyte leakage; RWC—relative water content.

**Table 4 plants-13-02912-t004:** Chemical attributes of the substrate used in the experiment.

pH	OM	P	K^+^	Ca^2+^	Mg^2+^	Na^+^	Al^3+^	H^+^ + Al^3+^	BS	CEC	ESP	ECse
H_2_O	%	mg dm^−3^	----------------------- (cmol_c_ dm^−3^) -------------------------								%	dS m^−1^
7.7	33.6	135.40	1.99	10.63	8.43	0.46	0.00	0.70	21.51	22.21	2.07	1.45

pH = hydrogen potential; OM = organic matter (wet oxidation—Walkey–Black); P = phosphorus (Mehlich 1), K^+^ = potassium (Mehlich 1), Ca^2+^ = calcium (KCl 1 M); Mg^2+^ = magnesium (KCl 1 M), Na^+^ = sodium (KCl 1 M); Al^3+^ = exchangeable acidity (KCl 1 M); H^+^ + Al^3+^ = potential acidity (calcium acetate 0.5 N); BS = sum of bases; CEC = cation exchange capacity; ESP = exchangeable sodium percentage; ECse = electrical conductivity of saturation extract.

## Data Availability

Data are contained within the article.

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
