# Peer review of "Potential of Ca-Complexed in Amino Acid in Attenuating Salt Stress in Sour Passion Fruit Seedlings"

_plants, 2024, doi:10.3390/plants13202912_

Round 1
Reviewer 1 Report (Previous Reviewer 2)
Comments and Suggestions for Authors
Dear authors and editor:
The manuscript has been considerably improved with respect to its initial version. I am thankful to the authors for having considered most of the corrections and/or modifications suggested in my former revision. Nevertheless, there are still several text and/or writing modifications that should be performed before considering it suitable for publication. Please see the pdf files attached for detailed comments and suggestions regarding these aspects.

Comments on the Quality of English LanguageMinor English Language editing is required, in my opinion., Nevertheless, other aspects not directly related with English editing have to be revised.
Author Response
Potential of complexed-Ca in attenuating salt stress in sour passion fruit seedlings
Cover letter
Dear Editor,
Thank you for forwarding the Reviewers' reports and comments on the manuscript ID (plants-3146058) currently under consideration. With that, I am sending the response to the concerns raised by the reviewers and the revised manuscript with all changes highlighted in red. For those suggestions/questions that were not addressed, a justification has been added to this letter.
Reviewer 1
According to what is indicated in lines 101-105, the experimental design comprises THREE factors (Passion fruit cultivar, salinity level and Ca source). As it is written here, it seems that the effect of cultivar and salinity levels have been considered independently from Ca source, and, on the other hand, the effect of the latter was analyzed without taking into account the effects of (and the putative interactions due to) the former two. If this is right, then I cannot understand how all the putative interactions among the different factors were evaluated...(please see also comments in the other PDF file attached). Please correct the writing since Table 2 (for example) shows that a THREE WAY ANOVA was indeed performed!
Answer: Reviewer position, very responsible for the suggestion. We oil and try to re-make it in a way that makes it more understandable. “The means for the cultivars, electrical conductivities of the water and, calcium sources”.
- instantaneous WUE (iWUE): use the same terminology in Tables and text. Please indicate in Materials and Methods how was it estimated.
Answer: Dear reviewer, thank you very much for the correction. In Table 2, the acronym for water use efficiency (WUE) was mistakenly written as QUE, and we left it in the text and figure as WUE, which is determined by the ratio between the photosynthetic rate (A) and leaf transpiration (E). We justify that we left the same acronym (WUE) because there is already a variable for intrinsic water use efficiency (iWUE), which is determined by the ratio between photosynthetic rate (A) and stomatal constancy (gs). In addition, we included how the variables were determined in the item: Material and Methods, as can be seen: “From these data, water use efficiency (WUE) was determined by the A/E ratio and intrinsic water use efficiency (iWUE) by the A/gs ratio.”
- I guess this stands for the triple interaction term, right?. Then please add : "(see triple interaction effect, Table 2)", at the end of the phrase.
Answer: Dear reviewer, thank you for the suggestion. The discussion in this section is related to the effect of calcium sources on the BRS SC1 cultivar irrigated with moderately saline water, in which there was no effect of calcium sources. We decided to merge the paragraph below with this one, since it is a continuation of the discussion. At the beginning of the discussion of Table 2, we discussed that water use efficiency responded to the triple interaction.
- This sentence should have been included in the previous paragraph.
Nevertheless, according to Table 2, only salinity treatment had a significant effect on iWUE, while all the remaining effects (including double interactions) should have been considered NS!
Answer: Dear reviewer, I made a correction in Table 2. The acronym for water use efficiency (WUE) was typed incorrectly (QUE), and has been corrected in Table 2. Thank you very much for your observation.
- This can be gathered in the same paragraph since responses of the same trait are being described.
Answer: Dear Reviewer, Suggestion accepted.
- Please, revise all the legends and avoid repeating "passion fruit" unnecessarily.
Answer: Dear Reviewer, Suggestion accepted.
- The writing of this sentence is not clear, please reformulate. The relationship between these two traits is just POSSIBLE, but not direct or categorical (see also comments on this subject in the other pdf file attached). The interpretation of changes in RWC should be cautious and considering the remaining responses analyzed…
Answer: Dear reviewer, thank you for the suggestion. We have decided to remove the sentence.
- The effect of the different treatments on ABA content and/or metabolism was not evaluated in the present research, hence this comment can be avoided.
Answer: The sentence has been removed.
- But given that stomatal conductivity was indeed affected by some treatments, then alterations in stomatal conductivity may also have contributed to the changes in iCO2. Please reformulate this explanation considering also this fact. (E.g.: Nevertheless, given that stomatal conductivity decreased under XXXX condition, its impact on Ci should not be discarded....) or so...
Answer: Dear reviewer, thank you for your suggestion. The suggestion was accepted and inserted into the text: “as well as by stomatal constraints (Figure 3C) that may limit CO2 efflux”.
Best regards, corresponding author.

Reviewer 2 Report (New Reviewer)
Comments and Suggestions for Authors
I am convinced that the manuscript fully meets the requirements of a highly authoritative journal and can be accepted for publication.
Author Response
Potential of complexed-Ca in attenuating salt stress in sour passion fruit seedlings
Cover letter
Dear Editor,
Reviewer 2
Dear reviewer, we appreciate your corrections/suggestions for improving the manuscript. We remain at your disposal to clarify any questions you may have.
Best regards, corresponding author.

Reviewer 3 Report (New Reviewer)
Comments and Suggestions for Authors
The study revealed that Ca-AA application attenuate salt stress in sour passion fruit. The physiological analysis is useful. However, some issues should be addressed before publication.
1. The major problem of this manuscript is that the results indicated that Ca-OA didn’t attenuate salt stress but Ca-AA did, which indicates that Ca2+ is not the major contributor here but amino acid is. Therefore, I suggested to change the title into “Potential of Ca Complexed in Amino Acid in Attenuating Salt Stress in Sour Passion Fruit Seedlings”.
2. Line 277, “the response to moderately water saline…” should be “moderately saline water”
3. Line129: The number ‘129’ is overlapped with ‘pH’ in the Table 1. The similar problem happened in Line 398-400.
4. Line 343, “On the other hand, the application of Ca-OA and Ca-AA resulted in the highest chlorophyll a index in the BRS GA1 and BRS SC1 cultivars, respectively”. The Figure 5 didn’t support this sentence.
Line 502: “The application of Ca-complexed with amino acids showed the damage caused by toxic salts in the photosynthetic activity and efficiency of seedlings (Figure 4)”. The sentence is confusing. First, Figure 4 is not about photosynthetic efficiency. Second, the application of Ca-AA showed the damage? Enhanced or decreased? Please be specific.
Comments on the Quality of English LanguageA few typoes require attention.
Author Response
Potential of complexed-Ca in attenuating salt stress in sour passion fruit seedlings
Cover letter
Dear Editor,
Thank you for forwarding the Reviewers' reports and comments on the manuscript ID (plants-3146058) currently under consideration. With that, I am sending the response to the concerns raised by the reviewers and the revised manuscript with all changes highlighted in red. For those suggestions/questions that were not addressed, a justification has been added to this letter.
Reviewer 3
- The major problem of this manuscript is that the results indicated that Ca-OA didn’t attenuate salt stress but Ca-AA did, which indicates that Ca2+ is not the major contributor here but amino acid is. Therefore, I suggested to change the title into “Potential of Ca Complexed in Amino Acid in Attenuating Salt Stress in Sour Passion Fruit Seedlings”.
Answer: Dear reviewer, we appreciate the suggestion and have modified the title of the paper as suggested.
- Line 277, “the response to moderately water saline…” should be “moderately saline water”.
Answer: Thank you. The suggestion was accepted.
- Line129: The number ‘129’ is overlapped with ‘pH’ in the Table 1. The similar problem happened in Line 398-400.
Answer: Dear reviewer, We tried to adjust, but always after line 128, Word inserts it to line 130. While line 398 goes to line 400.
- Line 343, “On the other hand, the application of Ca-OA and Ca-AA resulted in the highest chlorophyll a index in the BRS GA1 and BRS SC1 cultivars, respectively”. The Figure 5 didn’t support this sentence.
Answer: Dear reviewer, Thank you for the suggestion. We have removed the sentence.
- Line 502: “The application of Ca-complexed with amino acids showed the damage caused by toxic salts in the photosynthetic activity and efficiency of seedlings (Figure 4)”. The sentence is confusing. First, Figure 4 is not about photosynthetic efficiency. Second, the application of Ca-AA showed the damage? Enhanced or decreased? Please be specific.
Answer: Dear reviewer, thank you for your suggestion. The sentence has been reworded and we have inserted the correct figure (Figure 2).
Best regards, corresponding author.

This manuscript is a resubmission of an earlier submission. The following is a list of the peer review reports and author responses from that submission.
Round 1
Reviewer 1 Report
Comments and Suggestions for Authors
The authors present a study with the roles of Ca-complexed sources in moderately saline water stress in sour passion fruit seedlings, which provides a basis for further functional analysis of the this stress. However,
1. The figures should be improved. For example, figure 1 can be improved due to the typeface and should be supplied as supplementary material.
2. This study should measure more stress-related parameters to further reveal whether calcium really attenuating salt stress in sour passion fruit seedlings. It would be even better if the expression levels of stress-related genes could be detected.
3. The design and results of this manuscript are relatively simple, and deeper experiments need to be added. For example, RNA-seq.
4. There were some grammar and typo mistakes present in this manuscript.
With regret, I think that this manuscript cannot be accepted for publication in Plants, and the authors should to try another more relevant journal.
Comments on the Quality of English LanguageExtensive editing of English language required.
Reviewer 2 Report
Comments and Suggestions for Authors
Dear authors and editor:
The main objective of the present research was to analyze the role of Ca-complexed sources in reducing the effects of saline water stress on physiological aspects (mainly related to photosynthetic activity, water balance and electrolite leakage) of sour passion fruit seedlings. The experiment was carried out in a randomized block design with a 2 × 2 × 3 factorial scheme, consisting of two cultivars of sour passion fruit (BRS 23 GA1 and BRS SC1), two levels of water salinity (electrical conductivity of 0.5 and 4.0 dS m-1) and three sources of complexed calcium (without complexed Ca; Ca complexed with organic acids or Ca complexed with amino acids). Plants were grown in polyethylene bags during 60 days after sowing, with the treatments being applied at 12 and 32 days after seedling emergence. Traits were measured at the end of the experimental period.
Although the paper is in general well written, several text corrections are necessary and the wording of some paragraphs, particularly in the results section, should be reformulated to facilitate data analysis. On the other hand, some key aspects regarding the statistical analyses performed should be clarified. With respect to the experimental design, the commercial Ca-complexed sourses used differ in the composition (%) of key plant macronutrients (mainly N, K and Mg), and also the solutions used to implement the saline water treatments had different CaCl2.2H2O concentrations. The authors neither highlight these differences while describing the methodology adopted, nor discuss their putative effects on the traits recorded.
Finally, in the Discussion section, the interpretation of some of the responses and/or the putative mechanisms involved, despite being based on published research works elsewhere, are highly speculative, and additional physiological traits should have been measured in the context of the present experimental design to properly support them.
Please find detailed comments on these and other aspects in the revised pdf file attached.

Please see comments to the editor and authors